# Adaptive Distribution Calibration for Few-Shot Learning with Hierarchical Optimal Transport

**Dandan Guo [1,2], Long Tian[3], He Zhao [4], Mingyuan Zhou[5], Hongyuan Zha[1,6]**

[1]School of Data Science, The Chinese University of Hong Kong, Shenzhen
[2] Institute of Robotics and Intelligent Manufacturing
[3]Xidian University    [4]CSIRO's Data61    [5]The University of Texas at Austin
[6] Shenzhen Institute of Artificial Intelligence and Robotics for Society

guodandan@cuhk.edu.cn   tianlong@xidian.edu.cn   he.zhao@ieee.org
mingyuan.zhou@mccombs.utexas.edu   zhahy@cuhk.edu.cn

## Abstract

Few-shot classification aims to learn a classifier to recognize unseen classes during training, where the learned model can easily become over-fitted based on the biased distribution formed by only a few training examples. A recent solution to this problem is calibrating the distribution of these few sample classes by transferring statistics from the base classes with sufficient examples, where how to decide the transfer weights from base classes to novel classes is the key. However, principled approaches for learning the transfer weights have not been carefully studied. To this end, we propose a novel distribution calibration method by learning the adaptive weight matrix between novel samples and base classes, which is built upon a hierarchical Optimal Transport (H-OT) framework. By minimizing the high-level OT distance between novel samples and base classes, we can view the learned transport plan as the adaptive weight information for transferring the statistics of base classes. The learning of the cost function between a base class and novel class in the high-level OT leads to the introduction of the low-level OT, which considers the weights of all the data samples in the base class. Experiments on standard benchmarks demonstrate that our proposed plug-and-play model outperforms competing approaches and owns desired cross-domain generalization ability, proving the effectiveness of the learned adaptive weights. [1]

## 1 Introduction

Deep learning models have become the regular ingredients for numerous computer vision tasks such as image classification [1, 2] and achieve state-of-the-art performance. However, the strong performance of deep neural networks typically relies on abundant labeled instances for training [3]. Considering the high cost of collecting and annotating a large amount of data, a major research effort is being dedicated to fields such as transfer learning [4] and domain adaptation [5]. As a trending research subject in the low data regime, few-shot classification aims to learn a model on the data from the base classes, so that the model can generalize well on the tasks sampled from the novel classes. Several lines of works have been proposed such as those based on meta-learning paradigms [3, 6–11] and those directly predicting the weights of the classifiers for novel classes [12, 13]. Recently, methods based on distribution calibration have gained increasing attention. As a representative example, Yang et al. [14] calibrate the feature distribution of the few-sample classes by transferring the statistics from the base classes and then utilize the sampled data to train a classifier

---

[1]If you are interested in our work, please see our code, which is available at https://github.com/ DandanGuo1993/Adaptive-Distribution-Calibration-for-Few-Shot-Learning-with-Hierarchical-Optimal-Transport.

36th Conference on Neural Information Processing Systems (NeurIPS 2022).

for novel classes. A unique advantage of distribution calibration methods over others is that they build on top of off-the-shelf pretrained feature extractors and do not finetune/re-train the feature extractor.

The key of distribution calibration methods is to select the corresponding base classes and transfer their statistics for the labeled samples in a novel task. Existing approaches in this line usually do so with heuristic or less adaptive solutions. Specifically, Yang et al. [14] use the average features of the samples as the representation of a base class and select the top-$k$ (e.g., $k = 2$) closest base classes based on the Euclidean distance between the features of a novel sample and a base class. Despite the effectiveness of Yang et al. [14], it is questionable whether the Euclidean distance is the proper metric to measure the closeness between a base class and a novel sample since viewing a novel sample and a base class as points in the same space may not be the best solution. Moreover, it is less sound to characterize a base class only by the unweighted average over all its samples, when measuring its closeness with the novel sample. Representing a base class in this way would completely ignore the fact that each sample of a base class may contribute to the classification boundary differently. Finally, it may also be less effective to treat each of the top-$k$ base classes equally as their contributions can also be different, not to mention the omission of the other base classes.

To this end, this work develops a more adaptive distribution calibration method leveraging optimal transport (OT), which is a powerful tool for measuring the cost in transporting the mass in one distribution to match another given a specific point-to-point cost function. First, we formulate a distribution $P$ over the base classes and a distribution $Q$ over the labeled samples from the novel classes. With such formulation, how to transfer the statistics from the base classes to the novel samples can be viewed as the OT problem between two distributions, denoted as the high-level OT. By solving the high-level OT, the learned transport plan can be used as the similarity or closeness between novel samples and base classes. Since the high-level OT requires specifying the cost function between one base class and one novel sample, we further introduce a low-level OT problem to learn this cost automatically, where we formulate a base class as a distribution over its samples. In this way, the similarity between a novel sample and a base class is no longer representing a base class by the unweighted average over all its samples and then using the Euclidean distance. In our method, the weights of the samples are considered in a principled way. In summary, the statistics of base classes can be better transferred to the novel samples for providing a more effective way to measure the similarity between them. Notably, even in the challenging cross-domain few-shot learning, our H-OT can still effectively transfer the statistics from the source domain to the target domain.

We can refer to this adaptive distribution calibration method as a novel hierarchical OT method (H-OT) for few-shot learning, which is applicable to a range of semi-supervised and supervised tasks, such as few-shot classification [9] and domain adaptation [5]. Our contributions are summarized as follows: (1) We develop a new distribution calibration method for few-shot learning, which can be built on top of an arbitrary pre-trained feature extractor for being implemented over the feature-level, without further costly fine-tuning. (2) We formulate the task of transferring statistics from base classes to novel classes in distribution calibration as the H-OT problem and tackle the task with a principled solution. (3) We apply our method to few-shot classification and also explore the cross-domain generalization ability. Experiments on standardized benchmarks demonstrate that introducing the H-OT into distribution calibration methods can learn adaptive weight matrix, paving a new way to transfer the statistics of base classes to novel samples.

## 2 Background

### 2.1 Optimal Transport Theory

Optimal Transport (OT) is a powerful tool for the comparison of probability distributions, which has been widely used in various machine learning problems, such as generative models [15], text analysis [16, 17], adversarial robustness [18], and imbalanced classification [19]. Here we limit our discussion to OT for discrete distributions and refer the reader to Peyré and Cuturi [20] for more details. Denote $p = \sum_{i=1}^{n} a_i \delta_{x_i}$ and $q = \sum_{j=1}^{m} b_j \delta_{y_j}$ as two $n$ and $m$ dimensional discrete probability distributions, respectively. In this case, $\boldsymbol{a} \in \Delta^n$ and $\boldsymbol{b} \in \Delta^m$, where $\Delta^m$ denotes the probability simplex of $\mathbb{R}^m$. The OT distance between $p$ and $q$ is defined as

$$\text{OT}(p, q) = \min_{\mathbf{T} \in \Pi(p,q)} \langle \mathbf{T}, \mathbf{C} \rangle, \tag{1}$$

where $\langle \cdot, \cdot \rangle$ denotes the Frobenius dot-product; $\mathbf{C} \in \mathbb{R}_{\geq 0}^{n \times m}$ is the transport cost function with element $C_{ij} = C(x_i, y_j)$; $\mathbf{T} \in \mathbb{R}_{>0}^{n \times m}$ denotes the doubly stochastic transport probability matrix such that $\Pi(p, q) := \{\mathbf{T} \mid \sum_i^n T_{ij} = b_j, \sum_j^m T_{ij} = a_i\}$, meaning that $\mathbf{T}$ has to be one of the joint distribution of $p$ and $q$. As directly optimising Equation (1) can be time-consuming for large-scale problems, the entropic regularization, $H = -\sum_{ij} T_{ij} \ln T_{ij}$, is introduced in Cuturi [21], resulting in the widely-used Sinkhorn algorithm for discrete OT problems with reduced complexity.

## 2.2 Few-Shot Classification

Following a typical few-shot learning problem, we divide the whole dataset with labeled examples into a base dataset $\mathbf{D}_{\text{base}}$ with $B$ base classes and a novel dataset $\mathbf{D}_{\text{novel}}$ with $N_{all}$ novel classes, each with a disjoint set of classes. To build a commonly-used $N$-way-$K$-shot task [8, 14], we randomly sample $N$ classes from $N_{all}$ novel classes, and in each class, we only pick $K$ (e.g., 1 or 5) samples for the support set $\mathcal{S} = \{(x_i, y_i)\}_{i=1}^{N \times K}$ to train or fine-tune the model and sample $q$ instances for the query set $\mathcal{Q} = \{(x_i, y_i)\}_{i=N \times K+1}^{N \times K + N \times q}$ to evaluate the model. By averaging the accuracy on the query set of multiple tasks from the novel dataset, we can evaluate the performance of a model.

## 2.3 Distribution Calibration for Few-Shot Classification

Distribution calibration [14] uses the statistics of base classes to estimate the statistics of novel samples in the support set and generate more samples. Specifically, for the $b$th base class, its samples are assumed to be generated from a Gaussian distribution, whose mean and covariance matrix are:

$$\mu_b = \frac{1}{J_b} \sum_{j=1}^{J_b} \boldsymbol{x}_j, \quad \boldsymbol{\Sigma}_b = \frac{1}{J_b - 1} \sum_{j=1}^{J_b} (\boldsymbol{x}_j - \boldsymbol{\mu}_b)(\boldsymbol{x}_j - \boldsymbol{\mu}_b)^{\mathrm{T}}, \quad (2)$$

where $b \in [1, B]$, $\boldsymbol{x}_j \in \mathbb{R}^V$ is the $V$-dimensional feature of sample $j$ extracted from the pre-trained feature encoder, $J_b$ the number of samples in class $b$, and $\{\mu_b, \boldsymbol{\Sigma}_b\}$ are the statistics of base class $b$.

The samples of a novel class are also assumed to be generated from a Gaussian distribution with mean $\boldsymbol{\mu}'$ and covariance $\boldsymbol{\Sigma}'$. As the novel class only has one or a few labeled samples, it is hard to accurately estimate $\boldsymbol{\mu}'$ and $\boldsymbol{\Sigma}'$. Thus, the key idea is to transfer the statistics of the base classes to calibrate the novel class's distribution. Once the distribution of the novel class is calibrated, we can generate more samples from it, which are useful for training a good classifier. As a result, how to effectively transfer the statistics from the base classes is critical to the success of distribution calibration-based methods for few-shot learning. Accordingly, Free-Lunch [14] designs a heuristic approach that calibrates the Gaussian parameters of a novel distribution with one data sample $\boldsymbol{x}$:

$$\boldsymbol{\mu}' = \frac{\sum_{i \in \text{topk}(\mathbb{S}_d)} \boldsymbol{\mu}_i + \tilde{\boldsymbol{x}}}{k + 1}, \quad \boldsymbol{\Sigma}' = \frac{\sum_{i \in \text{topk}(\mathbb{S}_d)} \boldsymbol{\Sigma}_i}{k} + \alpha, \quad (3)$$

where: 1) $\tilde{\boldsymbol{x}}$ is the transformed data by Tukey's Ladder of Powers transformation (TLPT) [22], i.e., $\tilde{\boldsymbol{x}} = \boldsymbol{x}^\lambda$ if $\lambda \neq 0$ and $\tilde{\boldsymbol{x}} = \log(\boldsymbol{x})$ if $\lambda = 0$, for reducing the skewness of distributions and make distributions more Gaussian-like. 2) $\mathbb{S}_d = \left\{ -\|\boldsymbol{\mu}_b - \tilde{\boldsymbol{x}}\|^2 \mid b = [1, B] \right\}$ is a distance set defined by the Euclidean distance between the transformed feature $\tilde{\boldsymbol{x}}$ of a novel sample in support set and the mean $\boldsymbol{\mu}_b$ of the base class $b$. 3) $\text{topk}(\mathbb{S}_d)$ is the operation that selects the top-$k$ closest base classes from the set $\mathbb{S}_d$. 4) $\alpha$ determines the degree of dispersion of features sampled from the calibrated distribution.

Although effective, Free-Lunch represents a base class by the unweighted average over all its samples when computing its closeness with a novel sample, which ignores the fact that each sample of a base class may contribute to the classification boundary differently. In addition, the Euclidean distance in the feature space may not well capture the relations between a base class and a novel sample. Moreover, each selected top-$k$ base class has an equal weight (i.e., $1/k$), which may not reflect the different contributions of the base classes and omit useful information in unselected base classes.

# 3 Our Proposed Model

## 3.1 Overall Method

In this work, we propose a novel adaptive distribution calibration framework, a holistic method for few-shot classification. Compared to the novel classes, which only have a limited number of labeled

samples, the base classes typically have a sufficient amount of data, allowing their statistics to be estimated more accurately. Due to the correlation between novel and base classes, it is reasonable to use the statistics of base classes to revise the distribution of the novel sample. Therefore, the key is how to transfer the statistics from the base classes to a novel class to achieve the best calibration results, which is the focus of this paper. Here, we develop the H-OT to learn the *Transport Plan* matrix between base classes and novel samples, where each element of the transport plan measures the importance of each base class for each novel sample and more relevant classes usually have a larger transport probability. Computing the high-level OT requires the specification of the cost function between one base class and one novel class, which leads to the introduction of a low-level OT problem. By viewing the learned transport plan as the adaptive weight matrix, we provide an elegant and principled way to transfer the statistics from the base classes to novel classes.

## 3.2 Hierarchical OT for Few-Shot Learning

Moving beyond the Free-Lunch method [14], which uses the Euclidean distance between a novel sample and a base class to decide their similarity and endow the chosen base classes with the equal importance, we aim to capture the correlations between the base class and novel samples at multiple levels and transfer the related statistics from base classes to novel samples. We learn the similarity by minimizing a high-level OT distance between base and novel classes and build the cost function used in high-level OT by further introducing a low-level OT distance. To formulate the task as the high-level OT problem, we model $P$ as a following discrete uniform distribution over $B$ base classes:

$$P = \sum\nolimits_{b=1}^{B} \frac{1}{B} \delta_{R_b}, \tag{4}$$

where $R_b$ represents the base class $b$ in the $V$-dimensional feature space, which will be introduced later. Taking the $N$-way-1-shot task as the example, where each novel class has one labeled sample $x$, we represent $Q$ as a discrete uniform distribution over $N$ novel classes from support set:

$$Q = \sum\nolimits_{n=1}^{N} \frac{1}{N} \delta_{\tilde{x}_n}, \tilde{x}_n \in \mathbb{R}^{V \times 1}, \tag{5}$$

where $\tilde{x}_n$ is the transformed feature from $x$ following Yang et al. [14] and detailed below Equation 3. The OT distance between $P$ and $Q$ is thus defined as $\text{OT}(P,Q) = \min_{\mathbf{T} \in \Pi(P,Q)} \langle \mathbf{T}, \mathbf{C} \rangle$. We adopt a regularised OT distance with an entropic constraint [21] and express the optimisation problem as:

$$\text{OT}_\epsilon(P,Q) \stackrel{\text{def.}}{=} \sum\nolimits_{b,n}^{B,N} C_{bn} T_{bn} - \epsilon \sum\nolimits_{b,n}^{B,N} -T_{bn} \ln T_{bn}, \tag{6}$$

where $\epsilon > 0$, $\mathbf{C} \in \mathbb{R}_{\geq 0}^{B \times N}$ is the transport cost matrix, and $C_{bn}$ indicates the cost between base class $b$ and novel sample $n$. Importantly, the transport probability matrix $\mathbf{T} \in \mathbb{R}_{>0}^{B \times N}$ should satisfy $\Pi(P,Q) := \left\{ \sum_n^N T_{bn} = \frac{1}{B}, \sum_b^B T_{bn} = \frac{1}{N} \right\}$ with element $T_{bn} = T(R_b, \tilde{x}_n)$, which denotes the transport probability between the $b$th base class and the $n$th novel sample and is an upper-bounded positive metric. Therefore, $T_{bn}$ provides a natural way to weight the importance of each base class which can be used as the class similarity matrix when calibrating the novel distribution. Hence, the transport plan is the main thing that we want to learn from the data.

To compute the OT in Equation 6, we need to define the cost function $\mathbf{C}$, which is the main parameter defining the transport distance between probability distributions and thus plays the paramount role in learning the optimal transport plan. In terms of the transport cost matrix, a naive method is to specify the $\mathbf{C}$ with Euclidean distance or cosine similarity between the feature space of the novel sample and mean of the features from the base classes, such as $C_{bn} = 1 - \cos(\tilde{x}_n, \mu_b)$. However, these manually chosen cost functions might have the limited ability to measure the transport cost between a base class and a novel sample. Besides, representing the base class only with the average of all features in class $b$ might ignore the contributions of different samples for this class. Hence the optimal transport plan for these cost functions might be inaccurate. To this end, we further introduce a low-level OT optimization problem to automatically learn the transport cost function $\mathbf{C}$ in (6). Specifically, we further treat each base class $b$ as an empirical distribution $R_b$ over the features within this class:

$$R_b = \sum\nolimits_{j=1}^{J_b} p_j^b \delta_{x_j^b}, \quad x_j^b \in \mathbb{R}^V, \tag{7}$$

where $p_j^b$ is the weight of data $\boldsymbol{x}_j^b$ to base class $b$ and captures the importance of this sample and will be described in short order. Specifically, we train a classifier parameterized by $\phi$ with the samples in the base classes, which predicts which base class a sample is in. The predicted probability of sample $j$ belonging to the base class $b$ is denoted by $s_j^b$ and then $[p_1^b, \dots, p_{J_b}^b]$ is obtained by normalizing the vector $[s_1^b, \dots, s_{J_b}^b]$ with the Softmax function. We further define the low-level OT distance between each distribution $R_b$ and distributions $Q$ with an entropic constraint as

$$\text{OT}_\epsilon(R_b, Q) \stackrel{\text{def.}}{=} \sum\nolimits_{n,j}^{N,J_b} D_{jn}^b M_{jn}^b - \epsilon \sum\nolimits_{n,j}^{N,J} -M_{jn}^b \ln M_{jn}^b, \tag{8}$$

where the transport probability matrix $\mathbf{M}^b \in \mathbb{R}_{>0}^{J_b \times N}$ should satisfy $\Pi(R_b, Q) := \left\{ \sum_j^{J_b} M_{jn}^b p_j^b = \frac{1}{N}, \sum_n^N M_{jn}^b \frac{1}{N} = p_j^b \right\}$. The element $D_{jn}^b$ of cost function $\mathbf{D}^b \in \mathbb{R}_{>0}^{J_b \times N}$ is the distance between the $n$-th novel sample and the $j$-th sample from base class $b$, where we naturally use the distance between their features, i.e., $\tilde{\boldsymbol{x}}_n$ and $\boldsymbol{x}_j^b$. Empirically, we find that the cosine similarity $D_{jn}^b = 1 - \cos(\tilde{\boldsymbol{x}}_n, \boldsymbol{x}_j^b)$ works well in practice, although other choices are possible.

Minimizing the low-level OT loss in Equation (8) can learn an optimal transport probability matrix $\mathbf{M}^b$ for $b \in B$, where $M_{jn}^b$ tells us the transport weight between the $n$th novel class and $j$th sample in $b$ base class. Back to the cost function $C_{bn}$ for the high-level OT, instead of using the manually chose cost functions, we further adopt the learned total transport cost between novel sample $\boldsymbol{x}_n$ and all samples from base class $b$ in the low-level OT to represent the cost function $C_{bn}$ in high-level OT:

$$C_{bn} = \sum\nolimits_{j=1}^{J_b} D_{jn}^b M_{jn}^b, \tag{9}$$

where $C_{bn}$ is fed into (6) for learning the transport plan between base classes and novel samples. Defined in this way $C_{bn}$ is an adaptive cost between base class $b$ and novel sample $n$ in support set, taking full advantage of all samples in a base class. With the proposed low-level OT distance, the distance between a novel sample and a base class is no longer representing a base class by the unweighted average over all its samples and then using the Euclidean distance. In our method, the weights of the samples are considered in a principled way.

## 3.3 Calibrating Distribution and Training Classifier

Once we obtain the transport plan matrix by minimizing the OT problem in (6), we can compute the statistics of novel samples by following Yang et al. [14]. For the $n$-th transformed feature $\tilde{\boldsymbol{x}}_n$ in the support set, we calibrate its mean and covariance as follows:

$$\mu_n' = \frac{N \sum_{b \in B} T_{bn} \mu_b + \tilde{\boldsymbol{x}}_n}{B + 1}, \quad \boldsymbol{\Sigma}_n' = \frac{N \sum_{i \in B} T_{bn} \boldsymbol{\Sigma}_b}{B} + \alpha, \tag{10}$$

where $\mu_b$ and $\boldsymbol{\Sigma}_b$ are the statistics of base class $b$ by (2), respectively; $T_{bn}$ provides an adaptive way to weight the importance of base class $b$ for novel sample $n$; $N$ is used to scale $T_{bn}$ since $\sum_{b \in B} T_{bn} = 1/N$; $\alpha$ is a hyper-parameter explained in Equation (3).

For $N$-way-$K$-shot task with $K > 1$, the $N$ in aforementioned Equations (6) and (8) should be replaced with $N * K$, and the distribution calibration in Equation (10) should be undertaken $K$ times for each novel class. Notably, this way can potentially achieve more diverse and accurate distribution estimation for avoiding the bias provided by one specific sample. Thus, for a class $y \in N$ in novel task, we denote the calibrated distribution with a set of statistics as

$$\mathbb{S}_y = \left\{ \left( \boldsymbol{\mu}_{n1}', \boldsymbol{\Sigma}_{n1}' \right), \dots, \left( \boldsymbol{\mu}_{nK}', \boldsymbol{\Sigma}_{nK}' \right) \right\}. \tag{11}$$

Based on set $\mathbb{S}_y$ for novel class $y$, we can sample from the calibrated Gaussian distributions to generate a set of feature vectors with label $y$: $\mathbb{D}_y = \{(x, y) \mid x \sim \mathcal{N}(\mu, \Sigma), \forall(\mu, \Sigma) \in \mathbb{S}^y\}$. Given the transformed support set $\tilde{\mathcal{S}}$ with TLPT and the generated features $\mathbb{D}_y$, we only need to train the classifier for a task $\mathcal{T}$ (under, N-way-K-shot) by minimizing the cross-entropy loss:

$$\ell = \sum_{(\boldsymbol{x}, y) \sim \tilde{\mathcal{S}} \cup \mathbb{D}_y, y \in \mathcal{Y}^\mathcal{T}} -\log \Pr(y \mid x; \theta), \tag{12}$$

where $\mathcal{Y}^\mathcal{T}$ is the set of classes for the task $\mathcal{T}$ and the classifier is parameterized by $\theta$. We describe our proposed framework in Algorithm 1.

---

**Algorithm 1** Workflow about our adaptive distribution calibration on few-shot learning.

---

**Require:** Datasets $\mathbf{D}_{\text{base}}$, $\mathbf{D}_{\text{novel}}$, pretrained feature extractor $f(\cdot)$, classifier $\phi$, classifier $\theta$, hyper-parameters

Extract the features of $\mathbf{D}_{\text{base}}$ and $\mathbf{D}_{\text{novel}}$ with $f(\cdot)$; compute $\{\mu_i, \mathbf{\Sigma}_i\}_{i=1}^{B}$ by (2);

Train a classifier $\phi$ with $\mathbf{D}_{\text{base}}$ to compute $p_j^b$; Represent base class $b$ as $R_b$ with (7);

**for** $t$ in number of tasks **do**

    Randomly choose $N$-way-$K$-shot task;

    Learn $\mathbf{M}$ with (8) and compute cost $C_{bn}$ with (9);

    Learn the transport plan matrix $\mathbf{T}$ with (6);

    Calibrate the statistics with (10) and generate the samples from (11);

    Train the $N$-way classifier $\theta$ with (12) and test the query set on classifier $\theta$.

**end for**

---

# 4 Related Work

Recently, many efforts have been devoted to exploring the idea of learning to learn or meta-learning to alleviate the few-shot challenge. One approach to this problem is the optimization-based meta-learning algorithms [6, 7], which aim to learn a single set of model parameters that can be adapted to individual tasks with a small number of gradient update steps. Some simple but effective algorithms based on metric learning are also developed, with the goal of "learning to compare." For example, MatchingNet [8] and ProtoNet [9] learned to classify samples by comparing the distance to the representatives of each class. Another line of algorithms is to compensate for the insufficient samples by learning to augment. Most methods augment the training set using Generative Adversarial Networks (GANs) [23] or autoencoders [24] to generate samples [25–27] or features [28, 29].

Recently, Free-Lunch [14] is proposed to calibrate the distribution of the novel samples using the statistics from base classes and augment the data from the calibrated distribution. Although distribution calibration belongs to the scope of data augmentation, it does not need to build any complex generative models or fine-tune the backbone. Therefore, distribution calibration can be considered as a new promising line, where both Free-Lunch and our model fall into. Different from Free-Lunch which adopts a heuristic method to choose base classes for novel samples, we develop a novel hierarchical OT to learn the cost function and similarity (transport plan) between base classes and novel samples, providing a general and adaptive distribution calibration framework for few-shot learning. A recent metric-based work of connecting few-shot learning with OT is DeepEMD [30], which decomposes an image into a set of local features and uses the optimal matching cost between two images to represent their similarity. Besides, the authors of [31] introduce a prototype-oriented OT (POT) framework for set-structured data and apply it to metric-based few-shot classification. However, we directly minimize the OT between base classes and novel samples to learn the transport plan, which serves as the adaptive weight matrix in distribution calibration. Although both using OT for few-shot learning, DeepEMD, POT and ours are totally different methods. The hierarchical topic transport distance (HOTT) [32] is developed to measure the distance between documents, where documents are modeled as distributions over topics (solved by a high-level OT), and topics are further modeled as distributions over words (solved by a low-level OT). Different from HOTT which captures the semantic difference between documents by leveraging OT, topic modeling, and word embeddings, we develop the H-OT to learn the adaptive weight matrix for improving distribution calibration in few-shot learning, a task distinct from document representation.

# 5 Experiments

## 5.1 Experimental Setup

**Datasets** We evaluate our proposed method on several standard few-shot classification datasets with different levels of granularity, including miniImageNet [33], tieredImageNet [34], CUB [35], and CIFAR-FS [36]. Here, miniImageNet, tieredImageNet, and CIFAR-FS have a broad range of classes including various animals and objects while CUB is a more fine-grained dataset that includes various species of birds. The **miniImageNet**, which is randomly chosen from ILSVRC-12 dataset [37], contains 100 diverse classes with 600 samples per class. The image size is $84 \times 84 \times 3$. Following

Table 1: Classification accuracy (%) on miniImageNet, tieredImagenet and CUB with 95% confidence intervals. ∘ indicates the results reported by authors and ⋆ is our implementation of Free-Lunch with their released code.

| Methods | Backbone | miniImageNet | | tieredImagenet | | CUB | |
|---|---|---|---|---|---|---|---|
| | | 5way1shot | 5way5shot | 5way1shot | 5way5shot | 5way1shot | 5way5shot |
| E3BM∘ [41] | ResNet25 | 64.3 ± n/a | 81.0 ± n/a | 70.0± n/a | 85.0± n/a | - | - |
| LEO∘ [42] | WRN28 | 61.76 ± 0.08 | 77.59 ± 0.12 | 66.33 ± 0.05 | 81.44 ± 0.09 | - | - |
| MAML [6] by [3] | ResNet18 | 49.61± 0.92 | 65.72 ± 0.77 | - | - | 69.96 ±1.01 | 82.70±0.65 |
| Baseline++ ∘ [3] | ResNet18 | 51.87 ± 0.77 | 75.68 ± 0.63 | - | - | 67.02±0.90 | 83.58±0.54 |
| Negative-Cosine∘ [43] | ResNet12 | 63.85 ± 0.81 | 81.57 ± 0.56 | - | - | 72.66±0.85 | 89.40±0.34 |
| DeepEMD∘ [30] | ResNet12 | 65.91 ± 0.82 | 82.41 ± 0.56 | 71.16 ± 0.87 | 86.03 ±0.58 | 75.65±0.83 | 88.69±0.50 |
| MatchNet [8] by [30] | ResNet12 | 63.08 ± 0.80 | 75.99 ± 0.60 | 68.50 ± 0.92 | 80.60 ± 0.71 | 71.87±0.85 | 85.08±0.57 |
| MatchNet [8] by [31] | ResNet34 | - | 68.32 ± 0.66 | - | - | - | 84.66±0.55 |
| MatchNet +POT by [31] | ResNet34 | - | 68.51 ± 0.64 | - | - | - | 85.50±0.66 |
| ProtoNet [9] by [30] | ResNet12 | 60.37 ± 0.83 | 78.02 ± 0.57 | 65.65±0.92 | 83.40±0.65 | 66.09±0.92 | 82.50±0.58 |
| ProtoNet [8] by [31] | ResNet34 | - | 73.99 ± 0.64 | - | - | - | 87.33±0.48 |
| ProtoNet +POT by [31] | ResNet34 | - | 75.15 ± 0.63 | - | - | - | 88.34±0.46 |
| Free-Lunch⋆ [14] | ResNet12 | 64.73± 0.44 | 81.15±0.42 | 71.40±0.31 | 85.56±0.40 | 74.13±0.86 | 87.91±0.45 |
| **Our H-OT** | ResNet12 | 65.63±0.32 | 82.87 ±0.43 | 73.71±0.26 | 87.46±0.35 | 76.28±0.40 | 89.87±0.36 |
| Free-Lunch∘ [14] | WRN28+RTloss | 68.57 ± 0.55 | 82.88± 0.42 | 75.10 ± 0.32⋆ | 88.42± 0.40 ⋆ | 79.56±0.87 | 90.67±0.35 |
| **Our H-OT** | WRN28+RTloss | **69.04 ± 0.29** | **84.36 ± 0.41** | **75.91±0.35** | **89.33±0.48** | **81.23±0.35** | **91.45±0.38** |

the previous work [33], we split the dataset into 64 base classes, 16 validation classes, and 20 novel classes. The **tieredImageNet** is a larger subset of ILSVRC-12 dataset [37]. It contains 608 classes sampled from a hierarchical category structure, where each class belongs to one of 34 higher-level categories from the high-level nodes in the ImageNet. The average number of images in each class is 1281. We adopt 351, 97, and 160 classes for training, validation, and test, respectively. The **CUB** is a fine-grained few-shot classification benchmark, which contains 200 different classes of birds with a total of 11,788 images, and we resize them as $84 \times 84 \times 3$. We follow previous works [3] and split the dataset into 100 base classes, 50 validation classes, and 50 novel classes. The **CIFAR-FS** dataset is a recently proposed few-shot image classification benchmark, consisting of all 100 classes from CIFAR-100 [38]. The classes are randomly split into 64, 16, and 20 for meta-training, meta-validation, and meta-testing, respectively. Each class contains 600 images of size $32 \times 32 \times 3$.

**Implementation Details** Since Free-Lunch is the SOTA distribution calibration method, we mainly compare with Free-Lunch. For miniImageNet and CUB, we directly use pre-extracted features from Free-Lunch's code [14], which are produced by WideResNet trained from scratch on the base classes by following Mangla et al. [39] to optimize a classification loss and a self-supervised rotation loss jointly, denoted as WRN28+RTLoss. Free-lunch did not share the features on tieredImageNet and CIFAR-FS, thus we have taken the pre-trained WRN28+RTloss backbone of Mangla et al. [39] to extract the features, where we use this same backbone to implement Free-lunch with its official code and we have explored and selected an appropriate $w$ for Free-lunch. To confirm the validity of our proposed method on the commonly-used backbone, we also take the ResNet12 as the example, which is trained from scratch by minimizing a standard classification loss (without the rotation loss). To make sure the log transform in TLPT is valid to its inputs, we need to guarantee the non-negativity of features. Therefore, we extract the features of given samples from the penultimate layer (with a ReLU activation function) of the feature extractor. To demonstrate the effectiveness of our method, we consider the simple Logistic Regression (LR) classifier for $\theta$ and $\phi$ in Algorithm 1, where the LR implementation of scikit-learn [40] with the default settings is adopted. We use the same hyper-parameter value for all datasets except for $\lambda$. Specifically, the number of generated features is 750, the $\epsilon$ in Sinkhorn algorithm is 0.01, the $\alpha$ in (10) is 0.21; and $\lambda$ is 0.5, 1, 1 and 0.8 for miniImageNet, tieredImageNet, CUB, and CIFAR-FS, respectively, selected by a grid search using the validation set. The maximum iteration number in Sinkhorn algorithm is set as 200. The reported results are the averaged classification accuracy over 10,000 tasks.

## 5.2 Evaluation with the Standard Setting

We now conduct experiments on the most common setting in few-shot classification, 1-shot and 5-shot classification, where the results of different models on miniImageNet, tieredImagenet, CUB and CIFAR-FS are shown in Tables 1 and 2. To validate the effectiveness of our proposed model, we compare it with three main groups of the few-shot learning method, including optimization-based, metric-based, and distribution calibration-based. There are several observations. First, we can find

Table 2: 5way1shot and 5way5shot classification accuracy (%) on CIFAR-FS with 95% confidence intervals. ∗ means the results cited from [44] and ÷ denotes our implementation of Free-Lunch with their official code.

| Methods | Backbone | CIFAR-FS | |
|---------|----------|----------|---|
| | | 5way1shot | 5way5shot |
| MAML∗ [6] | 32-32-32-32 | $58.9 \pm 1.9$ | $71.5 \pm 1.0$ |
| ProtoNet∗ [9] | 64-64-64-64 | $55.5 \pm 0.7$ | $72.0 \pm 0.6$ |
| RelaNet∗ [45] | 64-96-128-256 | $55.0 \pm 1.0$ | $69.3 \pm 0.8$ |
| R2D2∗ [46] | 96-192-384-512 | $65.3 \pm 0.2$ | $79.4 \pm 0.1$ |
| Shot-Free∗ [47] | ResNet12 | $69.2 \pm$ n/a | $84.7 \pm$ n/a |
| MetaOpt∗ [48] | ResNet12 | $72.6 \pm 0.7$ | $84.3 \pm 0.5$ |
| RFS∗ [44] | ResNet12 | $73.9 \pm 0.8$ | $86.9 \pm 0.5$ |
| Free-Lunch÷ [14] | ResNet12 | $73.8 \pm 0.4$ | $86.0 \pm 0.4$ |
| Our H-OT | ResNet12 | $74.5 \pm 0.4$ | $87.1 \pm 0.3$ |
| Free-Lunch [14] | WRN28+RTloss | $74.9 \pm 0.4$ | $86.2 \pm 0.5$ |
| Our H-OT | WRN28+RTloss | $\mathbf{75.4 \pm 0.3}$ | $\mathbf{87.5 \pm 0.3}$ |

that our H-OT equipped with WRN28+RTLoss and simple linear classifier outperforms all previous competing methods on 5-way-1-shot and 5-way-5-shot settings, which illustrates the superiority of the proposed framework. Second, our H-OT equipped with ResNet12 is better than other baselines except for the 5way1shot task on miniImageNet, where in this case our H-OT is still competitive with DeepEMD [30]. This observation proves the effectiveness of distribution calibration and transferring the statistics from base classes to novel classes using OT. Third, the comparison between our model and the strong baseline of distribution calibration-based methods, i.e., Free-Lunch [14], confirms the validity of incorporating the adaptive weight matrix when transferring the statistics from the base classes to the novel classes. Although both methods utilize distribution calibration to perform few-shot classification, our model can more effectively transfer the knowledge of the base classes to estimate the distribution of novel samples more accurately with the help of hierarchical OT (H-OT). Fourth, in terms of 5-way 1-shot tasks on all datasets, the performance gain from our model over Free-Lunch on the CUB is larger than that of other datasets. However, for 5-way 5-shot tasks on all datasets, the improvement gain from our model over Free-Lunch on the CUB is not as significant as that on other datasets. The reason behind this might be that CUB is a fine-grained image classification dataset, which makes it more effective for statistics transfer compared to other datasets under the 1-shot setting. Furthermore, with the development of label examples in the novel classes (5-shot), our method can perform more effective knowledge transfer from base classes to novel samples in the coarse-grained image classification dataset. The results reveal that the granularity of the dataset is an important factor in the current few-shot classification setting. Our model avoids the costly fine-tuning for backbone and it only takes time at the testing stage. We defer the details on comparison of computational cost to Appendix A and summary of test results to Appendix B.

## 5.3 Evaluation with the Cross-domain Setting

To explore the cross-domain generalization ability of H-OT for the few-shot classification task, we consider a practical evaluation setting following Chen et al. [3], where there exists domain shift between base and novel classes (e.g., sampling base classes from a coarse-grained dataset and novel classes from a fine-grained dataset). We design two cross-domain scenarios: miniImageNet → CUB, CIFAR-FS → CUB, where CUB is the target domain while both miniImageNet and CIFAR-FS are the source domains. Compared with collecting images from a general class, collecting images from fine-grained classes might be more difficult, so these two scenarios are very meaningful in reality. We consider the comparison with Free-Lunch, which is closely related to ours. As shown in Fig. 1, both H-OT and Free-Lunch achieve acceptable performance in these two scenarios, proving the effectiveness of the distribution calibration framework in the cross-domain case. Besides, H-OT shows its clear advantage over Free-Lunch. It indicates that H-OT can

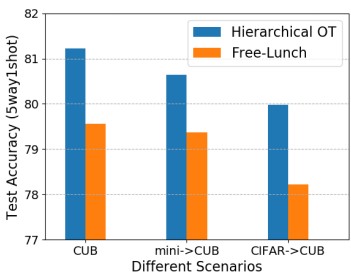

Figure 1: 5way1shot accuracy under the cross-domain scenarios, i.e., miniImageNet → CUB and CIFAR-FS → CUB. CUB indicates that both base and novel classes are from CUB.

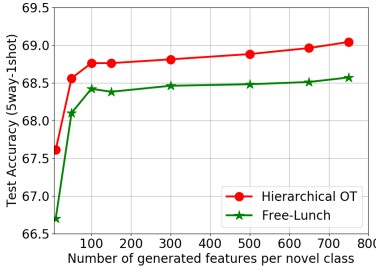

Figure 2: Accuracy on miniImageNet for 5way1shot task with varying number of generated features.

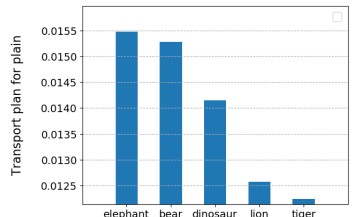

Figure 3: Top-5 closest base classes for a novel sample from the "plain" class on CIFAR-FS based on our learned transport plan.

better utilize the statistics from the base classes of the source domain with the adaptive weight matrix and transfer them to the novel classes of the target domain. Moreover, H-OT under the CIFAR-FS → CUB (blue bar in the 3rd column) scenario still outperforms Free-Lunch under CUB (orange bar in the 1st column). These results show that H-OT is less affected by domain shift between the base and novel classes and possesses the desired cross-domain generalization ability.

Table 3: Ablation study with different cost functions (5way1shot).

| Models | miniImageNet | CUB | CIFAR-FS | tieredImagenet |
|---|---|---|---|---|
| Free-lunch [14] | $68.57 \pm 0.55$ | $79.56 \pm 0.87$ | $74.94 \pm 0.64$ | $75.10\pm0.32$ |
| High-level OT+Euc | $68.79 \pm 0.31$ | $80.29 \pm 0.25$ | $75.09 \pm 0.33$ | $75.31\pm0.36$ |
| High-level OT+Euc+weighted | $68.80 \pm 0.32$ | $80.33\pm 0.27$ | $75.13\pm 0.31$ | $75.35\pm0.40$ |
| High-level OT+cos | $68.82 \pm 0.36$ | $81.01 \pm 0.41$ | $75.12 \pm 0.31$ | $75.52\pm0.36$ |
| High-level OT+cos+weighted | $68.78 \pm 0.33$ | $81.02\pm 0.30$ | $75.10 \pm 0.34$ | $75.55\pm0.41$ |
| Our H-OT | $\mathbf{69.04 \pm 0.29}$ | $\mathbf{81.23 \pm 0.35}$ | $\mathbf{75.42 \pm 0.32}$ | $\mathbf{75.91\pm0.35}$ |

Table 4: 5way1shot performance on miniImageNet with different backbones.

| Backbone | Free-Lunch | Ours |
|---|---|---|
| Conv4 | $54.62 \pm 0.64$ | $\mathbf{55.23 \pm 0.59}$ |
| Conv6 | $57.14 \pm 0.45$ | $\mathbf{58.01 \pm 0.42}$ |
| ResNet10 | $64.41 \pm0.33$ | $\mathbf{65.22 \pm 0.37}$ |
| Resnet18 | $61.50 \pm 0.47$ | $\mathbf{62.61 \pm 0.41}$ |
| WRN28+RTloss | $68.57 \pm 0.55$ | $\mathbf{69.04 \pm 0.29}$ |

## 5.4 Ablation Study and Qualitative Analysis

**Impact of adaptive cost from low-level OT** Recall that we formulate our general goal as a high-level OT problem and we propose a low-level OT for computing the cost function for the high-level one in an adaptive way. The low-level OT captures the distance between a base class and a novel class. Here we consider two commonly-used cost functions to replace the adaptive cost, including Euclidean distance and cosine similarity between the feature space of the novel classes and the mean of the features from the base classes, denoted as High-level OT+Euc and High-level OT+cos. Besides, to exclude the possibility that the gain of our H-OT comes only from the effect of introducing weighted mean in the low-level OT, we also consider use the weighted mean with (7) to implement High-level OT, denoted as High-level OT+Euc+weighted and High-level OT+cos+weighted. As summarized in Table 6, our proposed H-OT achieves better performance than these degraded variants, which confirms the validity of adaptive cost from low-level OT in few-shot classification. And introducing the weighted mean into the cost function of high-level OT does not improve the performance. It indicates that the performance gain comes mainly from the low-level OT rather than the effect of weighted mean. The reason behind this might be that our proposed low-level OT considers all samples within a base class when deciding the cost between novel samples and this base class, instead of using the mean of the features from the base class. Besides, even with the given cost functions, our proposed method still outperforms the competing baseline (i.e., Free-Lunch), indicating the usefulness of the transport plan (i.e., adaptive matrix) learned from high-level OT.

**Different backbones** Since our framework is agnostic with the choice of backbones, following Yang et al. [14], we further employ other commonly used backbones, such as networks with four or six convolutional layers (Conv4 or Conv6), Resnet10 and Resnet18, which are trained from scratch following [3]. Table 4 illustrates the classification performance (5-way-1-shot) comparison between our H-OT and strong baseline (Free-Lunch) on miniImageNet with different backbones. We find that H-OT always outperforms its baseline given the same backbone. This result reveals the superiority of learning an adaptive weight matrix to select statistics of base classes on different backbones. As H-OT is agnostic to the feature extractor and does not require costly fine-tuning, it can also conveniently work with more advanced feature extractors to get better performance.

**Number of generated features** Fig. 2 shows the variation of the classification performance of distribution calibration models with the increasing number of generated features, where we consider our H-OT (red) and Free-Lunch (green) in a 5-way-1-shot classification setting for miniImageNet. We can see that increasing the number of generated samples results in consistent improvement in both cases. However, given the same number of generated samples, H-OT outperforms its baseline, which only selects the $top - k$ base classes for each novel sample using Euclidean distance [14]. Interestingly, our model achieves around 68.5 accuracy with only 50 generated samples, while Free-Lunch takes 750 generated samples to achieve a similar result. Hence, the adaptive weight matrix learned by H-OT can contribute to the distribution calibration strategy for sampling more representative features.

**Learned transport plan** To explore whether the learned transport plan can capture the correlations between the base classes and one given sample from the novel class ("plain"), we show a list of base classes with decreasing ranks in Fig. 3. The top-5 base classes are highly related to the given novel sample about their semantic meanings. It is reasonable since the images from wild animal-related classes are usually taken on the plain, such as "elephant" and "bear". It indicates that the transport plan can be successfully optimized by our H-OT and provides an elegant way to measure the closeness between base classes and novel samples. Different from Free-Lunch that only selects top-$k$ base classes ($k$=2), our model takes full advantage of all base classes for learning adaptive weight for each base class. Additional quantitative results and qualitative are deferred to the Appendix B.

## 6    Conclusion

This paper introduces a novel hierarchical OT to improve the existing distribution calibration-based algorithm for few-shot learning. We first develop a low-level OT problem to learn the cost between novel samples and a base class, which takes into account different weights of the samples in the base class. Moreover, we further design a high-level OT for measuring the distance between novel samples and all base classes, which uses the cost learned from the low-level one to define the cost function rather than manually chosen cost functions. The proposed H-OT can capture the correlations between the novel samples and base classes with a two-level hierarchy. When transferring the statistics from base classes to novel samples, we view the learned optimal transport plan between them as the adaptive weight matrix, providing an effective way to weigh each base class for a novel sample. Extensive experiments have been conducted, showing that our proposed framework achieves competing performance on commonly few-shot learning problems and cross-domain scenarios.

## 7    Negative Societal Impacts

We develop a simple and effective distribution calibration approach to the few-shot learning, which has the potential to encourage researchers to derive new and better methods for few-shot learning or meta-learning. Our work may indirectly lead to a negative outcome if there is a sufficiently malicious or ill-informed choice of a few-shot classification task.

**Acknowledgements.** This work is partially supported by a grant from the Shenzhen Science and Technology Program (JCYJ20210324120011032) and Shenzhen Institute of Artificial Intelligence and Robotics for Society.

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
