# A   Computational Complexity

To approximate the general optimal transport (OT) distance between two discrete distributions of size $n$, the time complexity bound scales as $n^2 \log(n)/\varepsilon^2$ to reach $\varepsilon$-accuracy with Sinkhorn's algorithm, as demonstrated by Chizat et al. [49], Altschuler et al. [50], and Dvurechensky et al. [51]. In this paper, we set the maximum iteration number as Itermax $= 200$ in Sinkhorn algorithm for all experiments, which is typically enough for Sinkhorn algorithm to converge. We compare the computational cost of these two algorithms at the testing stage, on a Pentium PC with 3.7-GHz CPU and 64 GB RAM. Notably, similar to Free-Lunch, our proposed model is built on top of an arbitrary pretrained feature extractor and can avoid costly fine-tuning. Thus, we compare the computational complexity of two algorithms at the testing stage in Table 5. Although our model has a higher computational cost (our limitation) than Free-Lunch for introducing the hierarchical OT, it produces better performance with an acceptable cost. Below, we develop a deeper analysis.

For the high-level OT, we approximate the OT distance between the discrete distribution $P$ and distribution $Q$, where the former has the size of $B$ for $B$ base classes, and the latter has the size of $N * K$ for the $N$-way-$K$-shot task. Now the time complexity bound scales as $O(\max(B, N * K)^2 \log(\max(B, N * K)))/\varepsilon^2$ to reach $\varepsilon$-accuracy with Sinkhorn's algorithm. For the low-level OT, we approximate the OT distance between the distribution $R_b$ from the $b$th base class with $J_b$ samples and distribution $Q$, where the former has the size of $J_b$ and the latter has the size of $N * K$ for the $N$-way-$K$-shot task. Notably, when the number of samples in the $b$th base class is especially much, we can use the sub-sampling to build the empirical distribution $R_b$ and then solve the low-level OT problem for saving the computational complexity. For high-level OT, $B$ is usually larger than $N * K$ and its time complexity is $O(B^2 \log(B)/\varepsilon^2$. For low-level OT, the $J_b$ is generally larger than $N * K$ and thus the time complexity of low-level OT bound scales as $O(J_b^2 \log(J_b)/\varepsilon^2$. Here, the number $B$ of base classes of miniImageNet, CIFAR-FS, CUB, and tieredImageNet is 64, 64, 100, and 351, respectively. Besides, the average number $J_b$ of images in each base class of miniImageNet, CIFAR-FS, CUB, and tieredImageNet is 600, 600, 59, and 1281, respectively. As we can see, tieredImageNet owns the largest $B$ and $J_b$, whose running time is $9.31s$ for 5way1shot and $41.12s$ for 5way5shot. Therefore, our model can produce better performance with an acceptable overhead on large-scale dataset with larger numbers of classes and images. Notably, our model avoids the costly fine-tuning for backbone and it only takes time at the testing stage.

# B   Additional results

Table 5: Comparison of computational cost when testing a novel task, where s denotes seconds.

| Dataset | Free-Lunch[14] | | Our Hierarchical OT | |
|---|---|---|---|---|
| | 5way1shot | 5way5shot | 5way1shot | 5way5shot |
| miniImageNet | 2.34s | 8.41s | 2.91s | 10.06s |
| tieredImageNet | 5.76s | 24.61s | 9.31 | 41.12s |
| CUB | 2.11s | 8.24s | 2.60s | 11.12s |
| CIFAR-FS | 2.22s | 8.52s | 3.06s | 9.72s |

## B.1   Summary of the test results

To explore how our proposed model improves its baseline (Free-Lunch), we perform the experiments on 10,000 tasks from CIFAR-FS dataset using these two distribution calibration models, where the statistics about the classification results are shown in Fig. 4. We can see that our proposed model can effectively reduce the number of tasks with classification rates of less than $60\%$. To be our best knowledge, those novel tasks performed poorly by few-shot learning methods usually have the relatively large domain differences with all base classes, where the importance of each base class for novel sample might be similar. Different from Free-lunch, which only selects topw base classes to estimate the distribution of novel sample and might omit some relevant information, we utilizes all base classes by introducing the adaptive weight information over all base classes for each novel sample. It indicates that our proposed H-OT can effectively enhance distribution calibration method when there is a big domain difference between base and novel classes.

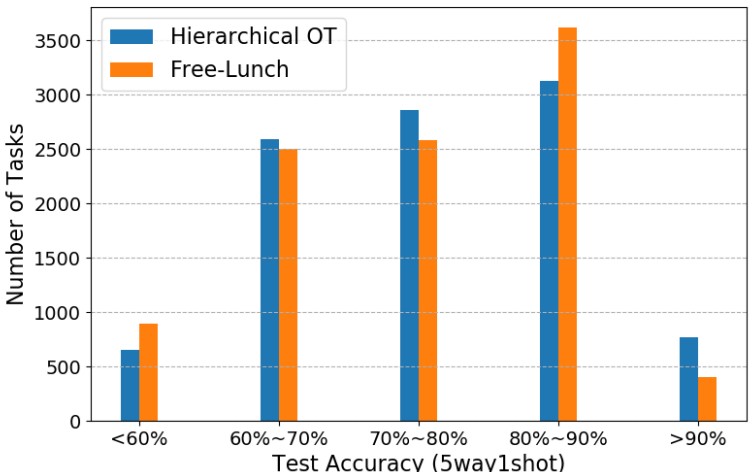

Figure 4: Statistics of the number of tasks among 10,000 tasks in total by their test accuracy.

## B.2 Ablation study about the number of retrieved base classes

To explore how the classification performance of distribution calibration models is influenced by the number of retrieved base classes, we compare Free-Lunch, our high-level OT+Euc, and our H-OT in a 5-way-1-shot classification setting for CUB, as summarized in Tab. 6. We consider 5 different settings to select the base classes, including top-1, top-2, top-5, top-10, and all. We find that increasing the number of retrieved base classes results in a sharp drop in performance in terms of Free-lunch, but gradually enhances the performance of our proposed methods. We attribute this to the learning of the adaptive weight for each base class in our method, which is more flexible and effective than Free-lunch.

Table 6: Ablation study with number of retrieved base class statistics (5way1shot on CUB).

| Models | top-1 | top-2 | top-5 | top-10 | all |
|---|---|---|---|---|---|
| Free-lunch | 78.37 ± 0.75 | **79.56 ± 0.87** | 78.97 ±0.80 | 78.49±0.91 | 69.14 ±0.88 |
| High-level OT+Euc | 80.08± 0.24 | 80.11 ± 0.23 | 80.15 ± 0.23 | 80.20±0.26 | **80.29 ± 0.25** |
| Our H-OT | 80.77 ±0.33 | 80.79 ± 0.30 | 80.87± 0.32 | 81.05 ±0.32 | **81.23 ± 0.35** |

## B.3 Visualization of generated features

To qualitatively show the effectiveness of our proposed methods, we show the t-SNE [52] representation of generated features by Free-Lunch and our proposed method in Fig. 5 and Fig. 6 given a same 5-way-1-shot task. We find that both our proposed model and Free-Lunch can generate more representative comprehensive features for each novel data. Besides, the generated features by ours are more discriminative than those by Free-Lunch. The phenomenon proves the benefits of learning the adaptive weight over base classes for novel sample.

## B.4 Learned weights on the toy dataset.

To intuitively reveal whether our proposed model can address the limitations of Free-Lunch and learn effective weights to measure base classes, we design a synthetic dataset with the following procedure.
1. We sample $B$ base classes from CIFAR-FS to build the base dataset. For easy observation, we set $B = 20$ and represent $\mu_b \in \mathbb{R}^V$ as the mean of the $b$th base class, and obtain the mean matrix $\mathbf{U} \in \mathbb{R}^{B \times V}$.
2. For the 5way1shot task, i.e., $N = 5$, $K = 1$, we use the gamma distribution with shape 0.8 and scale 1 to sample $N$ vectors and normalize each vector. Now we get a weight matrix $\mathbf{W} \in \mathbb{R}_{\geq 0}^{N \times B}$, where $\boldsymbol{w}_n \in \Delta^B$ is the probability simplex of $\mathbb{R}^B$.

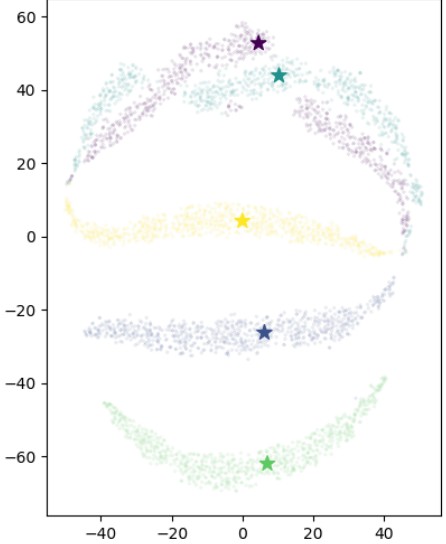
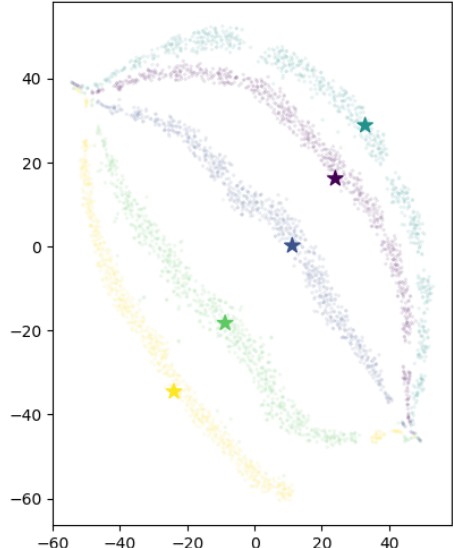

Figure 5: t-SNE projection of the generated features of Free-Lunch, where different colors represent different classes, the starts are the support set features and the small diamond are the generated features from the calibrated distributions, respectively.

Figure 6: t-SNE projection of the generated features of our H-OT, where different colors represent different classes, the starts are the support set features and the small diamond are the generated features from the calibrated distributions, respectively.

3. Finally, we synthesize $N$ novel samples by weighting the mean matrix $\mathbf{U}$ with $\mathbf{W}$, i.e., $\mathbf{X} = \mathbf{U}\mathbf{W}, \boldsymbol{x}_n \in \mathbb{R}^V$.

Given the base classes and 5way1shot task built from the generated novel samples, we adopt Free-Lunch and our high-level OT to learn the weight matrix, respectively, where the latter specifies the cost function with Euclidean distance between the novel sample and mean $\mu_b$ for a fair comparison. As shown in Fig. 7 in our revised version, we visualize the learned weights of different methods for a selected novel sample. Clearly, Free-Lunch does a hard selection of the top-2 base classes for the novel sample based on the Euclidean distance. However, the weights of the top-2 base classes are equal and the other base classes are ignored although some of them are also closely related to this novel sample. We observe that the weights based on our high-level OT can fit the ground-truth well. It indicates that our method can learn effective transport plan to weight the base classes for the novel sample.

## B.5 Learned transport plan matrix and adaptive cost based on Hierarchical OT

One of the benefits of developing the hierarchical optimal transport for the adaptive distribution calibration is the enhancement of model interpretability. To further examine whether our proposed H-OT can capture the correlations between the base classes and novel samples, in Fig. 8, we visualize the adaptive cost learned from low-level OT and the transport plan learned from the high-level OT given the adaptive cost on CIFAR-FS for 5way1shot task. It is clear that the adaptive cost function can effectively reflect the distance between base classes and novel samples. For example, the novel sample from class "baby" is closely related with the base classes, such as "house" , "couch", "castle". And the novel sample from class "plain" is closely related with "elephant", "dinosaur", "kangaroo", "bear", "lion", "tiger", "wolf" and others. Benefiting from the cost function, the transport plan learned from the high-level OT can measure the importance of each base class for each novel sample in a more adaptive way. We note that the transport probability between a base class and a novel class is usually larger if their cost is smaller (*i.e.*, more relevant). Therefore, it is reasonable to view the learned transport probability matrix as the adaptive weight matrix, which can reflect the different contributions of the base classes. These observations suggest that our proposed H-OT can produce adaptive cost function and the adaptive weight matrix, providing an elegant and principled way to transfer the statistics from base classes to the novel classes.

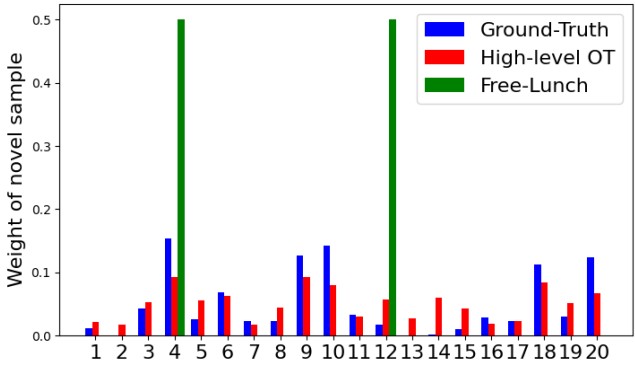

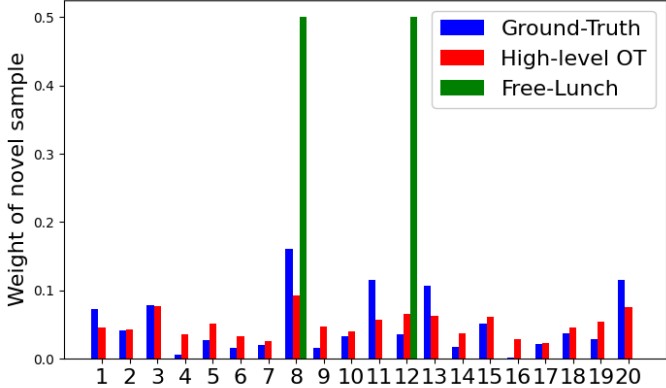

Figure 7: Examples of learned weights of different methods for two synthetic novel samples, respectively.

## B.6 Learned transport plan based on low-level OT

Taking the 5way1shot task on CIFAR-FS as an example, we further explore the per-example weights learned by our low-level OT. Specifically, we select the $b$th base class and the $n$th novel sample, and we visualize the learned $J_b$-dimensional weight vector $\{M_{j,n}^b\}_{j=1}^{J_b}$ in Fig. 9, where $J_b$ denotes the number of samples within the $b$th base class. As expected, $M_{j,n}^b$, which tells us the transport weight between the $n$th novel sample and $j$th sample in the $b$th base class, is very different among $J_b$ examples within the same base class. That is to say, each sample within the $b$th base class contributes to the $n$th novel class differently. Different from Free-lunch that measures the distance between novel sample $n$ and base class $b$ by only characterizing the base class as the unweighted average over all its samples, we use the learned weight vector to adaptively compute the distance between novel sample $n$ and base class $b$.

## B.7 Learned transport plan on the cross-domain setting

To further explore the learned transport plan when there is a significant shift between base and novel classes, in Fig. 10, we consider in-domain setting (CUB, top) and cross-domain setting (CIFAR-FS $\rightarrow$ CUB, bottom), both of which adopt same 5way1shot novel task from CUB. Compared with the in-domain scenario, we find that the transport plan learned in the cross-domain setting has a smaller magnitude of change. The reason might be that the distance between a base class and a novel class under the cross-domain settings is generally larger than that of the in-domain setting, making the importance of each base class for the novel sample more similar and reducing the gap between weights for base classes.

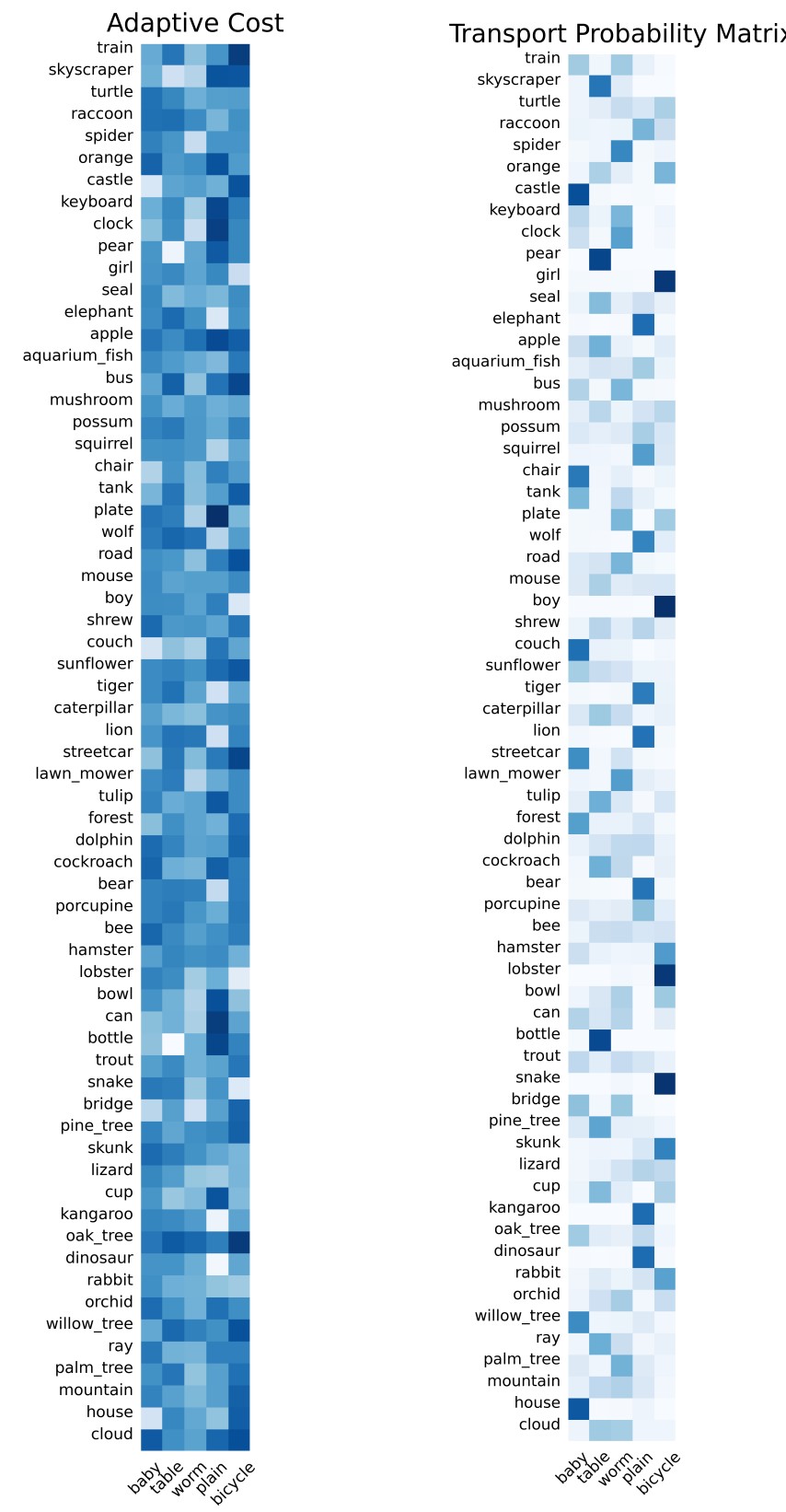

Figure 8: Learned cost between the base classes and novel samples and the resultant transport probability matrix, where we randomly select a novel task (5way1shot) from CIFAR-FS. The lighter the color is, the smaller the value is.

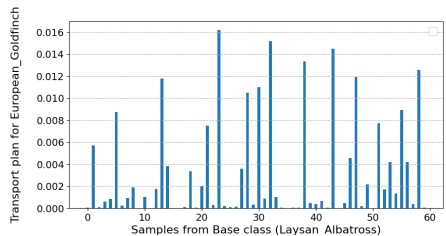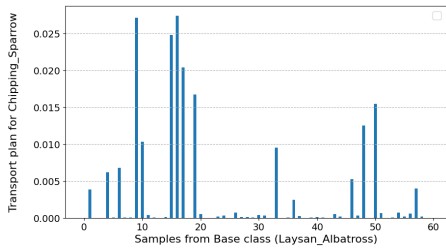

Figure 9: Learned per-example weights within base class Laysan Albatross, where the novel sample in the left figure is from the European Goldfinch class and novel sample in the right figure is from the Chipping Sparrow class.

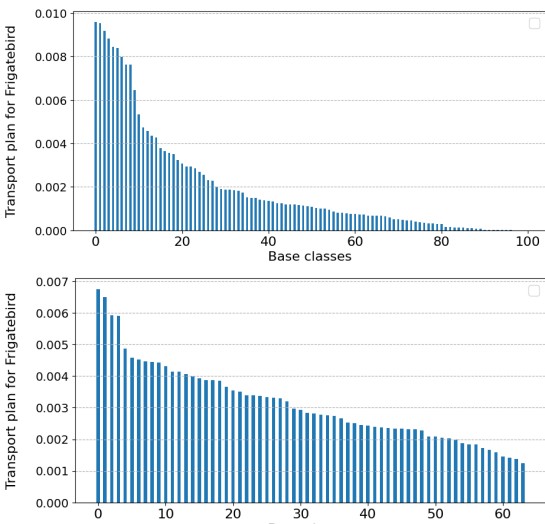

Figure 10: Learned transport plan for same 5way1shot task on CUB (only a novel class is presented for simplify). The base classes in top figure (in-domain) and bottom figure (cross-domain) are from CUB and CIFAR-FS, respectively.