# OpenReview forum: "Adaptive Distribution Calibration for Few-Shot Learning with Hierarchical Optimal Transport"
_NeurIPS.cc/2022/Conference — NeurIPS 2022 Accept_

### Official Review · Reviewer_aD8j · 2022-07-11

**Rating:** 6
**Confidence:** 3
**Soundness:** 3 good
**Presentation:** 2 fair
**Contribution:** 3 good

**Summary:**

This paper is on the topic of distribution calibration methods for few-shot learning, which seek to align the novel class feature distribution to that of the base classes. Methodological and experimental comparisons are primarily with a prior distribution calibration method (Free Lunch [14]) used the average features of the two closest base classes in Euclidean space. The authors instead propose using optimal transport to compare novel classes with all base classes. Additionally, the cost function to compare a base class with a novel sample is also learned as an optimal transport problem. Experiments are done on 5-way 1-shot and 5-way 5-shot miniImageNet, TieredImageNet, CUB, and CIFAR-FS, in comparison with Free Lunch and other popular few-shot methods. Overall, results are impressive, establishing a new state-of-the-art.

**Questions:**

1. Line 263: Why must the features be nonnegative?
2. How does the number of base classes affect this method? I presume that a higher number of classes scales the computational cost of this OT-based approach significantly, in a way that previous approaches (e.g. ProtoNet) do not.
3. Free Lunch [14] found that increasing the number of top-k base classes decreased performance, while the proposed method is somewhat akin to k=B and yet sees improvement. Is the difference due to the low-level OT method?


**Limitations:**

Discussion of limitations absent. What is the added computation cost of the method? When does OT tend to fail? How limiting is the Gaussian assumption of features?

**Strengths And Weaknesses:**

Strengths:
1. The proposed usage of optimal transport to align base and novel distributions is very sensible, as are the improvements over the previous distribution alignment method Free Lunch. It’s almost surprising that such an approach hasn’t been attempted before.
2. Background concepts (e.g. few-shot learning, optimal transport) are introduced well for unfamiliar readers, though some of the writing could use some improvement (see below), and the authors should take care to avoid text that is too similar to prior works (see below).
3. The empirical results are impressive. The main results are on 5-way 1-shot and 5-way 5-shot miniImageNet, TieredImageNet, CUB, and CIFAR-FS, which are the common FSL benchmarks in recent works. While all comparisons aren’t on the same backbone, the backbones used are listed in Table 1. Regardless, I believe these are the highest accuracies I’ve seen on these few-shot benchmarks, and the improvements over the most relevant baseline are consistent. Additional experiments in cross-domain settings and ablation studies lend further support to the efficacy of the method.

Weaknesses:
1. There isn’t a discussion on computational cost of this method. Optimal transport has a reputation for being expensive, so such a discussion would be very helpful, particularly given the two-level OT used here. How long does this method take to compute the transport plan?
2. Similar to the above point, the computational cost of the optimal transport plan scales with the number of classes involved. While this may be manageable for the small size of common FSL benchmarks, I suspect that datasets with larger numbers of classes may prove more expensive.
3. This work seems heavily based on the “Free Lunch” paper [14], not just in methodology, but also the paper itself. Some similarities are perhaps natural, as this work takes a similar strategy and builds off of [14]. However, Section 2.2 is mostly a paraphrase of [14]’s Section 3.1, there are strong similarities in certain lines of the Related Works, and several of the tables have the same formatting. There are also some odd variable substitutions from those of [14] that seem unnecessary (e.g. topw instead of the more common top-k, \beta for Tukey’s Ladder of Powers transformation instead of the more common \lambda).
4. Writing: While generally understandable, the writing isn’t always idiomatically or grammatically correct. I strongly recommend another thorough round of edits. A non-exhaustive list of examples appear under “Miscellaneous” below.

Miscellaneous:
- Line 21: [1] is a strange choice as the only cited reference for image classification.
- Line 30: “introduce to calibrate”
- Line 38: “Although with reasonably good performance”
- Line 51 (and elsewhere): “labelled” vs “labeled” <= Both spellings are correct, but generally one should stick to a single version.
- Line 75: “Owing a rich theory” <= “Due to its rich theory”? Though the “rich theory” isn’t the reason why people use OT for comparing probability distributions.
- Line 87: “We divide whole dataset” <= missing “the”
- Line 89/98: B is being used to denote both the number of base classes and the set of base classes.
- Line 99: “to generate from” => “to be generated from”
- Line 167: While I see why using only the mean may be inaccurate, I fail to see why it’d be “severely biased”.
- Line 175: “each distributions”
- Line 176: “should subject to”
- Line 214: incorrect usage of “Besides”
- Algorithm 1: “t” should be in math mode. “test the query set on classifier \theta” => “test classifier \theta on the query set”
- Line 238, 269: missing Oxford comma.
- Datasets shouldn’t be preceded with the word “the” unless followed by the word “dataset”
- Table 1: Why aren’t the best results on CUB bolded?
- Line 269: “CIFAR-FAS”
- Line 341: Starting a sentence with “And”
- [2] and [5] in the references are the same paper.

---

> ### Author Response · Authors · 2022-08-02
> **Response to Reviewer aD8j**
>
> Weakness 1 and weakness 2: Please see the response to Reviewers brZM and aD8j about the computational cost.
> Weakness 3 and weakness 4: Thanks for your suggestions. We have rewritten Section 2.2, adopted the common $\lambda$ for Tukey’s Ladder of Powers transformation and top-k, and corrected some typos in the revised version. We will further improve the presentation in our camera ready.
>
> Q1:  To make sure that the log transform in Tukey's Ladder of Powers transformation (TLPT) is valid to its inputs, we need to guarantee the non-negativity of features. Therefore, we extract the features of the given samples from the penultimate layer (with a ReLU activation function) of the feature extractor. We have explained the reason in our revision.
>
> Q2: Please see the response to Reviewers brZM and aD8j about the computational cost.
>
> Q3: To explore how the classification performance of distribution calibration models is influenced by the number of retrieved base classes, we compare Free-Lunch, our high-level OT+Euc, and our H-OT in a 5-way-1-shot classification setting for CUB. Tab. [6] in the Appendix of the revised version summarizes the results. A simple description about the results is as follows: We consider 5 different settings to select the base classes, including top-1, top-2, top-5, top-10, and all. The classification performance  of Free-Lunch under 5 settings is  78.37, 79.56 , 78.97, 78.49, and 69.14, respectively. However, our high-level OT+Euc achieves the performance of 80.08, 80.11, 80.15, 80.20, and 80.29; our hierarchical OT (H-OT) achieves the performance of 80.77, 80.79, 80.87, 81.05, and 81.23. We find that increasing the number of retrieved base classes results in a sharp drop in performance in terms of Free-lunch, but gradually enhances the performance of our proposed methods. We attribute this to the learning of the adaptive weight for each base class in our method, which is more flexible and effective than Free-lunch.

---

> > ### Comment · Reviewer_aD8j · 2022-08-07
> > **Response to Author Comments**
> >
> > I thank the authors for their responses. I’ve looked at the other reviews and the new revisions to the draft. I appreciate the authors have made many of the suggested corrections, though as I stated in my original review, my list was non-exhaustive, so I still recommend the authors carefully review the draft for writing (some more suggestions below that I encountered while reviewing the changes).
> >
> > I generally agree with many of the other reviewers’ comments in that this work builds heavily off Free Lunch. Also, reading the authors’ responses and looking at the timing results in the Appendix, it’s apparent that the computational cost associated with the proposed approach is indeed quite high. The extra cost relative to Free Lunch becomes especially apparent with increasing number of base classes. For example, 41.12 seconds for 5-way 5-shot TieredImageNet is quite slow, and real-world applications could easily have higher B or N, meaning even slower performance. Given that computational cost was also brought up in other reviews and can be a significant bottleneck for this method, I highly recommend including a deeper analysis, and not to say “We will leave the computational cost as a future study.” in the Appendix.
> >
> > Still, I feel there is enough interesting methodological contribution here that would be of interest to the community, and though the improvements over Free Lunch are relatively small, they are consistent and on mature few-shot benchmarks where gains are often incremental. I thus increase my score up to a 6, conditioned on the suggested improvements.
> >
> > Minor:
> > - Figure 1 caption: “5way 1 shot” not consistent with “5way1shot” throughout the paper (though I’d actually prefer 5-way 1-shot).
> > - Line 358: “Free lunch” vs “Free Lunch”
> > - Line 363: Broken reference to a Figure
> > - Line 624, 627: N and K should be in math mode.
> > - Line 674: “dinasour” => “dinosaur”
> > - Figure 7: The x-axis does not represent a series with any meaning. This is better shown as a bar plot.
> > - Figure 8 in the Appendix is grainy. Please update with a higher resolution version.
> > - I suggest moving up Table 2 and Figure 1 a line or two, for aesthetic reasons.
> > - The Negative Societal Impacts (Broader Impacts) should have been in the main paper, not the Appendix. The recent increase in the NeurIPS page limit to 9 pages was largely to accommodate these additions.

---

> > > ### Author Response · Authors · 2022-08-09
> > > **Response to Reviewer aD8j**
> > >
> > > Thanks for your careful checks and constructive comments that are valuable to us. As suggested, we have provided analytical and empirical studies in the revised appendix to understand the computational complexity better. Specifically, we provide a deeper analysis of the computational complexity in the blue texts of A, and the empirical running time is shown in Table 5 of the appendix. Moreover, we have corrected the typos and changed the figures as suggested in the recent revision.

---

### Official Review · Reviewer_pJGr · 2022-07-12

**Rating:** 7
**Confidence:** 4
**Soundness:** 4 excellent
**Presentation:** 2 fair
**Contribution:** 2 fair

**Summary:**

This work studies few-shot learning and proposes a optimal-transport-based weighting scheme to leverage examples from base classes for updating the classifiers of the learning of novel classes. Built on the framework of previous work [14], this work develops a optimal-transport-based weighting scheme instead of euclidean distance, and the resulting weights per base class implicitly considers similarity of individual examples within that base class to examples in the novel classes. Evaluation shows improved performance over [14] and competitive performance on standard few-shot image classification benchmarks across multiple backbones. Extensive ablation studies are conducted to verify superiority to [14].

**Questions:**

See weakness.

**Limitations:**

Limitations are sufficiently addressed.

**Strengths And Weaknesses:**

Strength

1. Extensive experiments across multiple backbones, cross-domain results and ablation studies. Qualitative results on optimal transport weights helps understanding of the approach.
2. Code provided for reproducing experiment results.

Weakness

1. The improvement over [14] in performance is relatively small with respect to how "principled" (L.59, L.191) the approach is. It would be nice to get an understanding on
- How much room of improvement is available from distribution calibration, to put the improvements that has been extracted by the proposed approach into perspective. For example, if there's oracle base class weighting, how good the performance can be on the query set?
- Are we moving in the right direction? For example, do the class weights (e.g. Appendix Figure 4) match the oracle weights? From [14] to this paper, are weights extracted looking more like oracle weights?

2. Would be nice to demonstrate how the per-example weights Mjn catches differences among examples in the same class. For example, baby is similar to sofa. Does Mjn tell apart sofa images with a baby and sofa images without a baby? How do the transport probabilities generalize in cross-domain settings?

======

The authors' response has adequately addressed my concerns. I have updated the score.

---

> ### Author Response · Authors · 2022-08-02
> **Response to Reviewer pJGr**
>
> Weakness 1: Please see the response to Reviewers Rd2f and pJGr about limitations of Free-Lunch, and whether the proposed method can address them and learn effective weights.
>
> Weakness 2: (1) Thanks for your suggestions. Taking the 5way1shot task on CIFAR-FS as an example, we further explore the per-example weights learned by our low-level OT. Specifically, we select the bth base class and the nth novel sample, and we visualize the learned {J_b}-dimensional weight vector \{M_{j,n}^{b}\}_{j=1}^{J_b}, where {J_b} denotes the number of samples within the bth base class. The results are visualized in Fig. [9] in Appendix of our revision. As expected, M_{j,n}^{b}, which tells us the transport weight between the nth novel sample and jth sample in the bth base class, is very different among J_b examples within the same base class. That is to say, each sample within the bth base class contributes to the nth novel class differently. Different from Free-lunch that measures the distance between novel sample n and base class b by only characterizing the base class as the unweighted average over all its samples, we use the learned weight vector to adaptively compute the distance between novel sample n and base class b.
>
> (2) To further explore the learned transport plan when there is a significant shift between base and novel classes, in Fig. [10] in Appendix of our revision, we consider in-domain setting (CUB, top figure) and above-mentioned cross-domain setting (from CIFAR-FS to CUB, bottom figure), both of which adopt the same 5way1shot novel task from CUB. Compared with in-domain scenario, we find that the transport plan learned in cross-domain setting has a smaller magnitude of change. The reason might be that the distance between a base class and a novel class under the cross-domain settings is generally larger than that of in-domain setting, making the importance of each base class for the novel sample more similar and reducing the gap between weights for base classes.

---

> > ### Comment · Reviewer_pJGr · 2022-08-09
> > **Response to author comments**
> >
> > Thanks for the answers to my questions. The answers adequately addressed my concerns.

---

> > > ### Author Response · Authors · 2022-08-09
> > > **Response to Reviewer pJGr**
> > >
> > > Thanks for your comments.

---

### Official Review · Reviewer_Rd2f · 2022-07-12

**Rating:** 5
**Confidence:** 2
**Soundness:** 3 good
**Presentation:** 2 fair
**Contribution:** 3 good

**Summary:**

This paper presents a calibration-based few-shot learning method to address the problem of the biased model toward novel samples. The main tool to that end is optimal transport (OT), and thereby this work attempts to overcome the limitations of its preliminary work [14] by replacing Euclidean distance with a proposed distance based on OT that can measure the distance between two distributions in a way of obtaining matching cost of one distribution to another. In order to get such a matching cost matrix, the paper suggests to learn those costs between base classes to novel samples by a low-level OT optimization problem, where the goal is to capture the importance of a sample with respect to its class. The experimental results show that the proposed method performs better than the existing few-shot learning methods including Free-Lunch [14].

**Questions:**

1. It would be better if any specific example is given to illustrate the problem of Euclidean distance is truly addressed by the H-OT distance.
2. The main strength of this work seems to lie in the effectiveness of cross-domain settings, and hence this part can be more emphasized throughout the paper.
3. In Sections 3.2-3.3, notations are too complicated and descriptions are not very effective. This part can hopefully be revised in a much simpler manner.

**Limitations:**

Yes.

**Strengths And Weaknesses:**

(Strengths)
1. [14] is a seminal work with clear novelty, and this work adequately finds room for improvement and indeed improves the performance of [14] by designing a more statistically sophisticated distance metric.
2. Experiments are well designed and conducted, and the proposed H-OT method seems to work well in most of the settings.
3. In particular, H-OT is highly effective in cross-domain settings, compared to Free-Lunch.

(Weaknesses)
1. There is no experiment to support the limitations of Free-Lunch as explicitly claimed by the authors. Even though it is observed that the overall performance has been improved, it is not quite clear whether such a performance gain indeed comes from addressing the limitations of Free-Lunch.
2. The presentation can be further improved as the current notations are too complicated and some consecutive sentences are not smoothly linked to each other.
3. In the overall performance comparison (Table 1), the performance gain of H-OT is somewhat marginal particularly in 1shot settings.

---

> ### Author Response · Authors · 2022-08-02
> **Response to Reviewer Rd2f**
>
> Weakness 1 and Question 1:  Please see the response to Reviewers Rd2f and pJGr about limitations of Free-Lunch, and whether the proposed method can address them and learn effective weights.
>
> Weakness 2 and Question 3:  We have identified and corrected some typos in the revised version. We will further improve the presentation in our camera ready.
>
>
> Weakness 3: Thanks for your comments. For 5way1shot settings, as listed in Tables 1 and 2, we can see that the performance gain of H-OT over Free-Lunch is about 0.47 on miniImageNet, 0.81 on tieredImagenet, 1.67 on CUB, and 0.5 on CIFAR-FS, respectively. It can be observed that the performance gain is very consistent. Especially, ours achieves more significant improvements on CUB. The reason behind this might be that CUB is a fine-grained image classification dataset, which makes it more effective for statistics transfer compared to other datasets under the 1-shot setting. We have added  an analysis of this phenomenon in the submitted version.
>
> Question 2:  We agree with you that our proposed method is very effective in cross-domain settings. Following your suggestion, we have emphasized this part throughout the paper in the revised version.

---

> > ### Comment · Reviewer_Rd2f · 2022-08-09
> > **Thank you for the responses.**
> >
> > Thank you for the responses to my comments. My major concern was about comparison with Free-Lunch, which is somewhat addressed and hence I will change my rating to 5 from 4.

---

> > > ### Author Response · Authors · 2022-08-09
> > > **Response to Reviewer Rd2f**
> > >
> > > Thanks a lot!

---

### Official Review · Reviewer_brZM · 2022-07-13

**Rating:** 6
**Confidence:** 4
**Soundness:** 3 good
**Presentation:** 3 good
**Contribution:** 3 good

**Summary:**

This paper proposes a novel few-shot classification algorithm based on distribution calibration. The proposed work is mainly based on Free-Lunch (Yang et al., 2021), where a feature extractor is first trained using base classes and the features of few-shot classes are augmented by the additional features drawn from Gaussian distributions. Here, the mean and covariance of the Gaussians are constructed by adequately adapting the statistics computed from the base classes, and Yang et al., 2021 proposed a simple heuristic approach for that. This paper suggests using a more theoretically grounded approach for this procedure, where the transfer of statistics from base classes is designed to be more sensitive to the importance of individual samples included in base classes or novel classes. Specifically, the paper introduces a hierarchical optimal transport algorithm where the relevance between the base and novel classes is estimated via solving hierarchical optimal transport problems and using the estimated transition matrices for constructing parameters for Gaussians to generate novel features.

**Questions:**

- How does the running time of the proposed algorithm (especially the time for solving the hierarchical OT problem) scale with the number of data ($N$ or $K$)? Does hierarchical OT incur a serious computational bottleneck?
- Can you notice a notable difference in the learned transport plan when there is a significant shift between base and novel classes? For instance, in the case of the cross-domain setting?
- Regarding the hierarchical optimal transport, I could find a relevant paper (Yurochkin et al., 2019) where the goal is to measure the distance between documents via hierarchical optimal transport; to better capture the semantic difference between documents, the paper proposes to first solve the lower-level topic space and use the learned topic transport for solving the higher-level document space optimal transport. I guess it is worth citing this paper and discussing its relevance.

**Limitations:**

As I mentioned before, most of the algorithm is based on Free-Lunch (Yang et al., 2021) and the only different part is the construction of Gaussian parameters through advanced weighted sums where the weights are computed from hierarchical optimal transport. Still, I think the contribution is quite valuable, and the algorithm demonstrated excellent performance.

**Strengths And Weaknesses:**

Strength
- The paper is well-written and easy to follow.
- The hierarchical optimal transport approach seems reasonable and sound.
- The experimental results are extensive and convincing. The paper also provides an ablation study and helpful visualizations.

Weakness
- The algorithm can be considered somewhat incremental because most of the procedure is based on (Yang et al., 2021).

---

> ### Author Response · Authors · 2022-08-02
> **Response to Reviewer brZM**
>
> Weakness: Thanks for your comments. Both our proposed method and Free-lunch fall into the distribution calibration group, which attracts increasing attention in few-show learning. For distribution calibration, how to decide the transfer weights from base classes with adequate samples to novel classes with a few samples is the key. Moving beyond that Free-lunch solves this problem with a heuristic approach, we formulate the task of transferring statistics from base classes to novel classes in distribution calibration as the H-OT problem and learn the adaptive weight matrix. Our proposed method produces the desired performance on standardized few-shot classification and cross-domain few-shot classification settings. Our H-OT paves a new way to transfer the statistics of base classes to novel samples, which provides a novel  insight for the future few-shot learning method.
>
> Q1: Please see the response to Reviewers brZM and aD8j about the computational cost.
>
> Q2: Thanks for your suggestions. To further explore the learned transport plan when there is a significant shift between base and novel classes, in Fig. [10] in Appendix of our revised version, we consider in-domain setting (CUB, top figure) and above-mentioned cross-domain setting (from CIFAR-FS to CUB, bottom figure), both of which adopt the same 5way1shot novel task from CUB. Compared with the in-domain scenario, we find that the transport plan learned in the cross-domain setting has a smaller magnitude of change. The reason might be that the distance between a base class and a novel class under the cross-domain settings is generally larger than that of the in-domain setting, making the importance of each base class for the novel sample more similar and reducing the gap between weights for base classes.
>
> Q3:  Thanks for bringing to our attention that reference, which we have now discussed in the related work section in the revised version. We agree that our general idea is conceptually related to Yurochkin et al. (2019), which propose to measure the distance between documents, where documents are modeled as distributions over topics (solved by high-level OT) and topics are further modeled as distributions over words (solved by low-level OT). However, ours studies a totally different problem/task. Moreover, Yurochkin et al. (2019) focus on the distance measure of two documents and make no use of the transport plan, while ours focuses on learning the transport plan for weighting the base classes and cares less about the actual distance.

---

> > ### Comment · Reviewer_brZM · 2022-08-07
> > **Thanks for the response**
> >
> > I appreciate the response by the authors which resolved some of my concerns. I keep my initial score.

---

> > > ### Author Response · Authors · 2022-08-09
> > > **Response to Reviewer brZM**
> > >
> > > Thanks for your comments.

---

### Author Response · Authors · 2022-08-02
**To all Reviewers**

We thank the reviewers for their constructive comments that are valuable for us to revise the paper. In the revised version, the changes to our manuscript are all highlighted in blue. In what follows we first address common concerns and then respond to each individual reviewer.

---

### Author Response · Authors · 2022-08-02
**Response to Reviewers brZM and aD8j about the computational cost:**

Thanks for your comments. We had reported the computational cost of our proposed method in the Appendix A.

More specifically, for the high-level OT, we approximate the OT distance between the discrete distribution P and distribution Q, where the former has the size of B for B base classes, and the latter has the size of N*K for the N-way-K-shot task. Now the time complexity bound scales as O(\max(B, N*K)^2 \log (\max(B,N*K)))/ \varepsilon^{2} to reach \varepsilon-accuracy with Sinkhorn's algorithm, as demonstrated by Chizat et al. [1], Altschuler et al. [2], Dvurechensky et al. [3]. For the low-level OT, we approximate the OT distance between the distribution R_b from the bth base class with J_b samples and distribution Q, where the former has the size of J_b and the latter has the size of N*K for the N-way-K-shot task. When the number of samples in the bth base class is especially much, we can use the sub-sampling to build the empirical distribution R_b and then solve the low-level OT problem for saving the computational complexity.


For high-level OT, B is usually larger than N*K and its time complexity is O(B^2 \log (B)/ \varepsilon^{2}. For low-level OT, the J_b is generally larger than N*K and thus the time complexity of low-level OT bound scales as O({J_b}^2 \log (J_b)/ \varepsilon^{2}. We had presented the running time of our proposed method and Free-Lunch on different datasets in Appendix, where the number of base classes (B) of miniImageNet, CIFAR-FS, CUB, and tieredImageNet is 64, 64, 100, and 351, respectively. Besides, The average number of images in each base class (R_b) of miniImageNet, CIFAR-FS, CUB, and tieredImageNet is 600, 600, 59, and 1281, respectively. As we can see, tieredImageNet owns the largest B and R_b, whose running time is 9.31s for 5way1shot and 41.12s for 5way5shot. Therefore, our model can produce better performance with an acceptable overhead on large-scale dataset with larger numbers of classes and images. Notably, our model avoids the costly fine-tuning for backbone and it only takes time at the testing stage.

---

### Author Response · Authors · 2022-08-02
**Response to Reviewers Rd2f and pJGr about limitations of Free-Lunch, and whether the proposed method can address them and learn effective weights:**

To intuitively reveal whether our proposed model can address the limitations of Free-Lunch and learn effective weights to measure base classes, we design a synthetic dataset with the following procedure.

1. We sample B base classes from CIFAR-FS to build the base dataset. For easy observation, we set B=20 and represent \mu_{b} \in  \mathbb{R}^{V} as the mean of the bth base class, and obtain the mean matrix \Umat \in  \mathbb{R}^{B \times V}.

2. For the 5way1shot task, i.e., N=5, K=1, we use the gamma distribution with shape 0.8 and scale 1 to sample N vectors and normalize each vector. Now we get a weight matrix \Wmat \in  \mathbb{R}_{\geq 0}^{ N \times B}, where \wv_n \in \Delta^{B} is the B-dimensional probability simplex.

3. Finally, we synthesize N novel samples by weighting the mean matrix with  weight matrix, i.e., \Xmat=\Umat \Wmat, \xv_n \in \mathbb{R}^{ V}.


Given the base classes and 5way1shot task built from the generated novel samples, we adopt Free-Lunch and our high-level OT to learn the weight matrix, respectively, where the latter specifies the cost function with Euclidean distance between the novel sample and mean  \mu_{b} for a fair comparison. As shown in Fig. [7] in Appendix of our revised version, we visualize the learned weights of different methods for a selected novel sample. Clearly, Free-Lunch does a hard selection of the top-2 base classes for the novel sample based on the Euclidean distance. However, the weights of the top-2 base classes are equal and the other base classes are ignored although some of them are also closely related to the novel sample. We observe that the weights based on our high-level OT can fit the ground-truth well. It indicates that our method can learn effective transport plan to weight the base classes for the novel sample.

---

### Meta-Review · Area_Chair_4Rks · 2022-08-27

**Recommendation:** Accept
**Confidence:** Certain

**Metareview:**

This paper builds off of the distribution calibration approach for few-shot learning known as “Free Lunch,” replacing the Euclidean metric with Hierarchical-Optimal Transport. The result is a more principled and empirically effective approach. The main issues among the reviewers were concerned with the limitations of Free Lunch and whether this new approach could overcome them, and what sort of additional computational cost would be incurred by using optimal transport. While more expensive, it is felt that overall the approach is sufficiently novel and effective.


**Award:**

No

---

### Decision · Program_Chairs · 2022-09-14

Accept